# Molecular Mechanisms of Anti-Neoplastic and Immune Stimulatory Properties of Oncolytic Newcastle Disease Virus

**DOI:** 10.3390/biomedicines10030562

**Published:** 2022-02-28

**Authors:** Volker Schirrmacher

**Affiliations:** Immune-Oncological Center Cologne (IOZK), D-50674 Cologne, Germany; v.schirrmacher@web.de

**Keywords:** autophagy, cancer immunotherapy, cancer vaccine, cancer therapy resistance, dendritic cell, exosome, immunogenic cell death, interferon-Ι, NK cell, T cell

## Abstract

Oncolytic viruses represent interesting anti-cancer agents with high tumor selectivity and immune stimulatory potential. The present review provides an update of the molecular mechanisms of the anti-neoplastic and immune stimulatory properties of the avian paramyxovirus, Newcastle Disease Virus (NDV). The anti-neoplastic activities of NDV include (i) the endocytic targeting of the GTPase Rac1 in Ras-transformed human tumorigenic cells; (ii) the switch from cellular protein to viral protein synthesis and the induction of autophagy mediated by viral nucleoprotein NP; (iii) the virus replication mediated by viral RNA polymerase (large protein (L), associated with phosphoprotein (P)); (iv) the facilitation of NDV spread in tumors via the membrane budding of the virus progeny with the help of matrix protein (M) and fusion protein (F); and (v) the oncolysis via apoptosis, necroptosis, pyroptosis, or ferroptosis associated with immunogenic cell death. A special property of this oncolytic virus consists of its potential for breaking therapy resistance in human cancer cells. Eight examples of this important property are presented and explained. In healthy human cells, NDV infection activates the RIG-MAVs immune signaling pathway and establishes an anti-viral state based on a strong and uninhibited interferon α,ß response. The review also describes the molecular determinants and mechanisms of the NDV-mediated immune stimulatory effects, in which the viral hemagglutinin-neuraminidase (HN) protein plays a prominent role. The six viral proteins provide oncolytic NDV with a special profile in the treatment of cancer.

## 1. Introduction

An oncolytic virus (OV) is defined as an agent that can selectively replicate in and kill cancer cells without damaging healthy cells. An OV can be an attenuated naturally existing virus, or it can be a genetically engineered recombinant virus. Four OVs are presently approved for some clinical cancer treatment: Oncorine (H101), Rigvir, T-VEC [1], and Delytact. The latter is a genetically engineered HSV-1 that was approved in Japan in 2021 for the treatment of patients with malignant glioma. Clinical trials with OVs are listed at www.clinicaltrials.gov (accessed on 28 June 2021).

Cassel et al. first reported the anti-neoplastic and immune stimulatory properties of oncolytic avian Newcastle Disease Virus (NDV) in 1965 [2]. Since this time, research on NDV has increased steadily up to the present day. Attenuated strains of this RNA paramyxovirus have been applied to cancer patients for more than 55 years, either as a single agent or in combination with a cancer vaccine. A brief history about NDV and its oncolytic strains was provided in 2000 [3]. Because of its oncolytic and pluripotent immune stimulatory properties, NDV has become an agent of worldwide research interest. New findings provide a deep insight into mechanisms of tumor-selective oncolysis and immune stimulation.

Basic information about the virus’s genome and its modes of cellular infection and replication are provided first before the molecular mechanisms of its intrinsic anti-neoplastic effects are described. The second part deals with the effect of NDV on normal healthy cells and its immune stimulatory capacities.

The milestone findings are listed in two tables. The summaries focus on the mechanisms of the anti-neoplastic and immune stimulatory effects, and the article highlights answers to important questions.

## 2. Basic Information on NDV

### 2.1. Genome

The genome of NDV is a non-segmented, negative-sense, single-stranded (ss) RNA of slightly less than 15200 nucleotides (nt). All the genome sizes of NDV are consistent with the “rule of six”, which is typical for Paramyxoviridae. This rule is thought to arise from the need to fully encapsidate the entire length of the genome with a chain of NP monomers that span exactly 6 nt. The nucleocapsides of the most prevalent genome-length NDV strains contain about 2530 NP monomers [4]. The genome is encapsidated by NP self-assembly into tightly packed helical 28 nm-diameter nucleocapsids with a clam-shaped structure which functions for genome transcription and replication [5,6].

The genomic RNA (3′-NP-P-M-F-HN-L-5′) contains a 55 nt 3′-extragenic untranslated region (3′UTR), known as Leader, and a 114 nt 5′-extragenic untranslated region (5′UTR) known as Trailer. These are control regions essential for transcription and replication and also for the encapsidation of newly synthesized RNAs into virus particles. Leader and Trailer flank the six genes that encode NP, phosphoprotein (P), matrix protein (M), fusion protein (F), hemagglutinin-neuraminidase protein (HN), and the large polymerase protein (L) [7].

### 2.2. Cellular Infection and Viral Replication

Infection of cells by NDV can occur via membrane fusion [4,5] or via endocytosis. In the case of endocytosis, NDV can use various pathways: clathrin-mediated endocytosis in non-lipid raft membrane domains [8], phagocytosis and macropinocytosis in mixed membrane domains [8], and RhoA-dependent endocytosis in lipid raft membrane domains [9].

It can schematically be divided into two steps:(i)Cellular infection. In the case of membrane fusion, cellular infection starts with the virus binding to a host cell’s surface with α 2,6-linked sialic acid from glycoproteins or glycolipids via the cell-adhesion domain of HN [10]. This is followed by the activation of F. The concerted action of HN and F leads to conformational change, enabling fusion of the viral and the host cell membrane and thereby opening a pore to deliver the viral genome into the cytoplasm [11].NDV can also enter cells via macropinocytosis or clathrin-mediated endocytosis, as studied in chicken embryonic fibroblasts [8]. RhoA-dependent endocytosis triggers the Rho GTPase Rac1 that impacts via p21-activated kinase 1 (Pak1) signaling, actin rearrangement, and plasma membrane ruffling [12]. NDV internalization can occur via macropinosomes and trafficking via Rab5a GTPase positive early endosomic vesicles [8]. The transfer of the viral genome from the vesicles to the cytoplasm occurs via membrane fusion [13]. The endocytosis pathway may release the nucleocapsid by accelerating the disintegration of the M proteins scaffold at acidic pH [13].In the cytoplasmic compartment, the negative-stranded RNA genome is transcribed stepwise into mRNAs and translated into proteins. This process involves double-stranded (ds) RNA intermediates. Transcription begins at a single promoter that is present in the Leader region. A triphosphorylated Leader RNA of 55 nt (ppp-RNA Leader) is synthesized and is neither capped nor polyadenylated and is not functional as mRNA. This is followed by the sequential transcription of the genes in the 3′-5′order via the RNA-dependent RNA polymerase activity of the L protein. The L polymerase-mediated transcription yields individual mRNAs by a stop–start mechanism guided by the conserved gene-start (GS) and gene-end (GE) signals. The mRNAs are also capped, methylated, and polyadenylated by L [4,7] and translated into proteins.The cap serves as a unique molecular module that recruits cellular proteins and mediates cap-related biological functions [14]. Cap-dependent translation begins with the recognition of the eucaryotic mRNA’s 7-methyl GTP (m7G) cap by eukaryotic translation initiation factor 4F (eIF4F). Many viruses compete with host cell mRNAs for translation factors and ribosomes.The three non-membrane proteins (NP, P, and L) are synthesized by ribosomes in the cytoplasm and the two membrane-associated proteins (HN, F) at the endoplasmic reticulum (ER) translocon complex [15]. M proteins are nuclear-cytoplasmic trafficking proteins that are positioned between the viral lipid membrane and the nucleocapsid [11].(ii)Replication. When a sufficient amount of NP protein accumulates in the cytoplasm, a switch can occur from RNA transcription to replication. The polymerase complex then ignores the transcription stop signals at the 3′end of each gene and a full-length, positive-sense antigenome is synthesized. These antigenomic replicative intermediates are totally encapsidated by viral NP monomers, just like the full-length, negative-strand genomic RNA/ NP complex.The phosphoprotein P is a non-catalytic subunit of the viral RNA polymerase. It serves as a bridge between the L protein and the NP-RNA template. The complex of P with unassembled NP monomers might be involved in the regulation of the switch from transcription to replication [4]. Two non-structural proteins, V and W, are expressed by mRNAs, which are derived from P via RNA editing. V inhibits the alpha/beta interferon response in bird cells [16] and is a determinant of host range restriction [17].The L protein is the largest structural protein and the least abundant in infected cells. It possesses all the enzymatic activities necessary for the synthesis of viral mRNAs and genome RNA replication, including nucleotide polymerization, mRNA capping, methylation, and polyadenylation. It only functions in association with P [18].

### 2.3. NDV Permissive and Non-Permissive Hosts

The permissive hosts of NDV are birds. NDV is widespread in many countries worldwide and can infect over 250 bird species. The non-permissive hosts of NDV are all vertebrates except birds.

Class II NDV strains are diverse, with at least 20 genotypes. They include most oncolytic strains of medical interest, e.g., lentogenic Ulster, B1 and La Sota, mesogenic Mukteswar and velogenic Italien, and Hert33.

## 3. Intrinsic Anti-Neoplastic Activities

Anticancer chemotherapeutic drugs with their relatively low level of tumor selectivity exert unwanted non-target side effects. New agents with higher tumor selectivity are therefore urgently needed. As oncolytic NDV has high tumor selectivity in humans, it is important to unravel the molecular mechanisms of this agent.

### 3.1. Targeting Rac1

The small Rho GTPase Rac1 is targeted by NDV in human Ras-transformed tumorigenic cell lines. In these cells, Ras is activated via membrane growth factor receptors (e.g., EGFR, PDGFR) [19]. Rac1 activates Pak1 which plays a central role in many oncogenic signaling pathways in human cells [20]. NDV interacts with Rac1 in transformed cells upon viral entry, during syncytium induction, and upon actin reorganization. This OV is therefore recommended for targeted therapy against the proliferation and invasion pathways of glioblastoma multiforme (GBM) [21]. Rac1 gene downregulation led to the inhibition of NDV replication [22].

### 3.2. Tumor-Selective Virus Replication

Oncolytic NDV can replicate up to 10,000-fold in human cancer cells [23]. It selectively kills tumor cells in vitro [23] and exerts antitumor effects in vivo in human tumor xenografts [24]. In contrast, oncolytic NDV does not replicate in and kill human non-transformed cells.

The reasons for this difference in susceptibility to NDV virus replication between tumorigenic and non-tumorigenic cells are multifold and will be discussed before going into the molecular details.

Cancer cells may downregulate key components of the type I interferon (IFN-I) signaling pathway, such as cytoplasmic protein kinase dsRNA activated (PKR) [25], retinoic acid-inducible gene I (RIG-I) [26,27], interferon regulatory factors (IRFs) [28], and cell surface IFN-I alpha receptor (IFNAR) [29]. Such defects of tumor cells are due to mutations in respective genes that allow escape from the growth inhibitory effect of IFN-I [30,31]. Tumor selective replication of NDV was found to be associated with a defective response of the antiviral enzymes PKR and myxovirus resistance A (MxA). While nontumorigenic human peripheral blood mononuclear cells reacted to UV-inactivated NDV with the induction of high levels of these enzymes, the tumor cells showed no response to UV-inactivated NDV [32].

In cancer cells, cell cycle regulation and cellular proliferation are typically disrupted. This has to do with the activity of oncogenes such as the small GTPase Ras and the downregulation of tumor suppressor genes such as those coding for retinoblastoma protein (Rb) [30]. Hyperactive Ras blocks PKR, which facilitates the tumor-selective replication of OVs [33]. Downregulated Rb diminishes cell cycle regulation. The cyclin-dependent kinase (CDK)-Rb-E2F axis forms the core transcriptional machinery driving cell cycle progression. Because of its importance, evolution has selected eight early 2 factor (E2F) genes for encoding transcriptional activators and repressors. Alterations in key components of the CDK-Rb-E2F axis occur in virtually all cancers and result in an increase in oncogenic E2F activity [33].

### 3.3. Tumor Selective Viral mRNA Translation

In Ras-transformed tumor cells, the p38 mitogen-activated protein kinase (MAPK) pathway (Ras/Raf/MEK/ERK) is activated by the binding of receptor ligands to the cognate receptor tyrosine kinase. With the help of adaptor proteins, the activated receptors convert Ras-GDP to Ras-GTP. The information is further transferred by phosphorylation (activation) of a series of enzymes (Raf, MEK1/2, ERK1/2), which leads to the activation of MAPK-interacting kinase 1 (Mnk1). Activated Mnk1 leads to dissociation of the 4EBP1-eIF4E complex, which results in the release, phosphorylation, and activation of the mRNA cap-binding eucaryotic translation initiation factor eIF4E [34]. The Mnk1/2-eIF4E axis is often dysregulated in cancer and recommended as a potential therapeutic target in melanoma [35].

The NDV infection of tumor cells activates the MAPK/Mnk1 pathways to upregulate the host cap-dependent translation machinery [35]. Interestingly, NP proteins were recently reported to interact with eIF4E to facilitate the selective translation of NDV viral mRNAs [36].

### 3.4. Tumor-Selective Shift to High Cytoplasmic and Cell Surface Expression of Viral Proteins

The NDV infection of human tumor cells, irradiated by 200 Gray (Gy), was demonstrated by FACS flow cytometry to lead to a shift towards the high-density cell surface expression of viral HN and F molecules [37]. Such shifts were not observed with nontumorigenic human cells.

Pronounced differences between human tumor cell lines and nontumorigenic cells were also observed upon infection by a recombinant lentogenic strain of NDV (Ulster) expressing an incorporated marker transgene (enhanced green-fluorescent protein, EGFP) (NDFL-EGFP). A long-lasting strong cytoplasmic EGFP fluorescence signal was observed in the whole cell population of tumor cells. In nontumorigenic cells, the fluorescence signals were only weak or missing completely [32].

### 3.5. Tumor-Selective Switch from Positive Strand RNA Translation to Negative Strand Antigenome Synthesis

A comparative analysis of human tumorigenic and nontumorigenic cells revealed that within 1 h of infection by NDV, the amount of positive-strand RNA increases in both cell types. In tumor lines, an increase in negative-strand RNA was observed after about 12 h. In contrast, in nontumorigenic cells the expression of negative-strand RNA remained weak and transitory [32].

Eukaryotic initiation factor 2α (eIF2α) phosphorylation exerts translational control in metabolism and the shut-off of cellular protein synthesis [38]. eIF2α phosphorylation was found to be higher in NDV-infected tumorigenic cells as compared to nontumorigenic cells [32]. It was hypothesized that the accumulation of phosphorylated eIF2α and the formation of typical stress granules (SGs), probably from the activation of cytoplasmic PKR and PKR-like endoplasmic reticulum kinase (PERK), occur at an early stage of NDV infection, while at a later stage the binding of increased NP protein to eIF4E enhances the shut-off of host protein translation [36]. Moreover, NP relocating to polysomes could contribute to selective translation of viral mRNAs [36].

The ppp-RNA Leader of NDV activates RIG-I [39] and PKR/eIf2α signaling activates antiviral SGs [40]. SGs are membraneless RNA-protein granules that assemble under various stress conditions and regulate mRNA translation and degradation. NDV-induced SGs contain T cell internal antigen (TIA)-1, Ras GTPase-activating protein-binding protein (G3BP)-1, eucaryotic initiation factors, and the small ribosomal subunit. Depletion of TIA-1 inhibited viral protein synthesis and reduced extracellular virus yields [40].

NDV infection also induces activation of the NLR family pyrin domain-containing 3 (NLRP3) inflammasome, which plays a role in pyroptosis [41]. This is an inflammatory form of cell death triggered by certain inflammasomes, leading to the cleavage of gasdermin D and the activation of inactive cytokines, such as IL-18 and Il-1ß [42].

In conclusion, normal cells prevent negative-strand antigenome synthesis due to the strong activation—via PKR, RIG-I, or Toll-like receptor (TLR) signaling [43]—of the anti-viral defense mechanisms. Tumor cells, in contrast, with their less-strong anti-viral defense machinery, enable oncolytic NDV antigenome synthesis, a prerequisite to viral replication.

### 3.6. Tumor-Selective Switch to Autophagy

More detailed information concerning NDV-mediated oncolysis became available within the last 15 years. This includes the unfolded protein ER stress response (UPR), UPR signaling, autophagy, and apoptosis [44].

ER plays an important role in regulating protein synthesis/processing, lipid synthesis, and calcium homeostasis. During virus infection, many viral proteins are synthesized by ER-associated ribosomes and transported into the ER lumen for proper folding or posttranslational modification. The chaperone Bip binds to unfolded/misfolded proteins and releases the ER stress sensors PERK, activating transcription factor 6 (ATF6), and inositol-requiring enzyme 1α (IRE1α), thus triggering UPR signaling. NDV was found to activate all three branches of UPR signaling: PERK-eIF2α, ATF6, and IRE1α. Knockdown of PERK or ATF6 inhibited NDV-induced autophagy and reduced the extent of NDV replication [44,45].

Autophagy [44] is an evolutionary conserved intracellular process. It generates a double-membrane vesicle to deliver portions of the cytoplasm to the lysosome for digestion and recycling. Autophagy can be initiated by diverse stress stimuli, including nutrient depletion, ER stress, oxidative stress, and virus infection. Remarkably, in 2016 it was reported that the ectopic expression of NP or P of NDV is sufficient to induce autophagy via the ER stress-related UPR response [44]. To recall, NP proteins protect the antigenome from RNA degradation and the viral proteins NP, P, and L are required for nucleocapsid assembly.

During autophagy, the antigenomic template of NDV becomes shielded from the cytoplasm. The catabolized proteins and lipids can be re-used by autophagy for anabolism in the process of virus production.

### 3.7. Tumor-Selective Oncolysis: Intrinsic Signaling Pathways

If ER homeostasis cannot be restored, UPR drives the damaged or infected cells to apoptosis [43]. eIF2α-CHOP-BcL-2/JNK and IRE1a-XBP1/JNK signaling was reported to promote apoptosis and inflammation and to support the proliferation of NDV [45]. BcL-2 is a mitochondrial anti-apoptotic protein. Knock down and overexpression studies showed that C/EBP-homologous protein (CHOP), IRE1α, X-box binding protein 1 (XBP1), and jun kinase (JNK) support efficient virus proliferation. The cellular translation shut-off caused by PERK/PKR-eIF2α signaling [32,45] and the hacking of the translational machinery by NDV infection via NP-eIF4E interaction [36] allow NDV-infected tumor cells to translate the viral proteins preferentially.

In addition to UPR signaling and autophagy, other factors, such as microRNAs, are also involved in tumor-selective oncolysis. MicroRNA-204 was significantly upregulated in NDV-treated human lung cancer cells. This was associated with caspase-3 and p38-MAPK signaling pathway-induced apoptosis and oncolysis [46]. Twenty-three microRNAs were predicted to target various genes involved in NDV replication and anti-viral immunity in HeLa cells, e.g., ErbB, JAK-STAT, NFκB, and RIG-I [47].

Oncolytic NDV was reported to the trigger cell death even of cancer stem cells. This was true for stem cell-enriched spheroids of lung cancer and inhibited the 3D growth potential in vitro [48]. NDV/FMW infection resulted in the degradation of microtubule-associated protein 1 light chain 3 (LC3) II and P62, two hallmarks of autophagy maturation. Apparently, NDV promoted autophagy flux in these spheroids. This was confirmed by transmission electron microscopy, revealing the appearance of an increased number of double-membrane vesicles [48].

Tumor-selective oncolysis relies on a crosstalk between OVs, UPR signaling, and autophagy [49]. This leads to viral proliferation and apoptosis. If apoptosis occurs too early after cell infection, there is not sufficient time for extensive virus replication. During evolution in birds, NDV has apparently adapted to cellular control mechanisms such as autophagy and mitophagy, which also function in human cancer cells. This is supported by the following findings: (i) NDV-induced autophagy and mitophagy promoted viral replication by blocking caspase-dependent apoptosis in lung cancer cells [50]; (ii) the PI3K/Akt pathway delayed cell death and increased cell survival as a means of improving virus replication [51]. Phosphoinositide-3-kinases (PI3Ks) and Rac GTPases are intracellular signal transducers able to regulate multiple signaling pathways fundamental for cell behavior. They are frequently dysregulated in human cancer and their existence crossroads between these pathways [52,53]. NDV seems capable of using multiple signaling pathways in tumor cells to facilitate viral replication.

The morphological hallmarks of NDV-induced apoptosis, as visualized by confocal microscopy images of infected HeLa cells, were: (i) active apoptosis with cellular shrinkage and DNA fragmentation, (ii) advanced apoptosis displaying extensive membrane blebbing, and (iii) late apoptosis showing apoptotic bodies [54]. NDV induces multimodal cell death responses according to programmed cell death, including apoptosis [37], autophagic cell death [44], necroptosis [37], pyroptosis [55], and ferroptosis [56].

### 3.8. Oncolysis-Independent Effects

Oncolysis-independent anti-tumor effects of NDV also exist. They have to do with improved adhesiveness between virus-infected tumor cells and cells from the immune system, such as NK and T cells. These immune stimulatory and co-stimulatory aspects will be reported below.

### 3.9. NDV Spread in Tumors

Following virus replication, NDV assembles and buds from the plasma membranes of the infected cells. In this process, M proteins play an important role. They have the potential to self-assemble into hollow helical oligomers [13] and to interact with viral NP and HN proteins and also with cellular multivesicular body (MVB) proteins [57,58].

The release of progeny virions from the surface of infected cells is facilitated by neuraminidase activity located at the sialidase ß-propeller domain [10] of the HN protein. This cleaves sialic acid from sugar side chains, thereby releasing the progeny virions [59].

An important virus protein for virus spread in tumors is F. Mutations at the cleavage site of the F precursor protein F_0_ facilitate multicyclic virus replication and represent an important virulence factor [60]. Syncytium formation was reported to occur in cell lines upon high-efficiency transfection, via a Semliki-Forest-Virus vector, of c-DNA coding for HN and F [61]. Single-stranded RNA viruses such as NDV and others are characterized by the highly efficient self-amplification of RNA in host cells and have been recommended as new types of RNA vaccines [62]. To make use of syncytium formation, a novel recombinant vesicular stomatitis virus (VSV) vector with incorporated HN and F genes from NDV was developed as a platform to treat cancer [63]. Interactions between HN and F not only induce syncytium formation but also trigger complete autophagy. This is mediated through the activation of the AMPK-mTORC1-ULK1 pathway [64]. AMPK is an energy-sensing adenosine monophosphate (AMP)-activated protein kinase, mTORC1 (mechanistic target of rapamycin complex 1) is a protein kinase, and ULK1 (unc-51-like autophagy activating kinase 1) is an autophagy-initiating protein kinase. Apparently, this network plays a role in autophagy and in maintaining cellular energy and nutrient homeostasis [64]. The formation of syncytia by fusogenic NDV strains finally leads to syncytium disintegration through apoptosis or necroptosis [55].

Viral NP proteins and microRNAs could be transferred to cells via NDV-related exosomes (NDV-Ex). They exhibited viral replication-promoting and IFN-ß-suppressing abilities [65,66]. NDV-Ex also enhanced NDV replication in chicken cells through exporting a distinct mRNA [67]. It is possible that similar mechanisms occur also in human tumors. Exosomes are nanosized vesicles released by the fusion of an organelle of the endocytic pathway, the MVB, with the plasma membrane [64]. They can encapsulate RNA, DNA, and proteins. Most likely, these incorporated molecules are involved in intercellular communications.

NDV taken up in Rab5a positive endosomes or macropinosomes [8] could in theory be recycled for export via exosomes. This would further facilitate viral spread in tumors. Interestingly, this hypothesis is supported by a recent report. Rab5a was found to be associated with genes involved in exosome secretion. By bioinformatics analysis, Rab5a, a member of the small Rab GTPase family, was found to interact with 37 genes involved in the exosome secretion regulatory pathways. Stable silencing of Rab5a could experimentally decrease exosome secretion [68].

NDV particles produced in autophagosomes could also be exported from the cell. Such a process, designated as secretory autophagy [69], could be relevant for viral spread in tumors. Microbial release from cells and transmission via secretory autophagy has been observed with a number of viruses (e.g., polio, morbillivirus, coxsackievirus, and influenza A virus) [69].

Viral spread in tumors has effects on the tumor microenvironment (TME). For instance, in dendritic cells (DCs) from the TME, NDV oncolysate (containing tumor-associated antigens (TAAs), NDV, exosomes, apoptotic bodies, pathogen- and damage-associated molecular patterns (PAMPs and DAMPs)) can be taken up by macropinocytosis and clathrin-mediated endocytosis. Material taken up by macropinocytosis is processed via the classical pathway for antigen presentation to CD4+ T cells. Material taken up by endocytosis is processed via sorting endosomes and late endosomes for antigen cross-presentation at the DC plasma membrane to CD8+ T cells [70]. In this way, oncolytic NDV in tumors could activate its TME and change it from “cold” (not immune cell infiltrated) to “hot” (immune cell infiltrated) areas.

### 3.10. Breaking Cancer Therapy Resistancies

So far, it was reported that the anti-neoplastic effects of NDV in humans include (i) targeting the oncogenic protein Rac1, (ii) replicating selectively in tumor cells via autophagy, (iii) selectively destroying tumor cells (viral oncolysis), and (iv) promoting virus spread via syncytia and exosomes. NDV can also suppress the glycolysis pathway, which is an important energy source for cell growth and proliferation [71].

Beyond these anti-neoplastic effects, oncolytic NDV has the intrinsic potential to break the resistance of cancer cells to a variety of therapies [27]. The main arguments and findings are as follows:(i)The breaking of resistance to therapies depending on cell proliferation, such as chemotherapy or radiotherapy. As NDV replicates in the cytoplasm of cells it is independent from DNA and cell replication. It has the potential to infect and replicate in non-proliferating cancer cells, such as irradiated cancer cells [72], cancer stem cells, or dormant tumor cells. A few examples of breaking drug resistance are: induction of apoptosis in cisplatin-resistant human lung adenocarcinoma cells [73], autophagy enhanced NDV oncolysis in drug-resistant lung cancer cells [74], and augmented growth-inhibiting and pro-apoptotic effects of temodal on glioblastoma cells in vitro and in vivo [75].(ii)The breaking of resistance to targeted therapy by small molecule inhibitors (e.g., anti-EGFR for lung cancer, anti-HER2/estrogen in breast cancer, and v-raf murine sarcoma viral oncogene homolog B (BRAF) protein inhibitors in melanoma). For example, under hyper-activation of Rac1, the Rac1-GTP activates Pak1, leading to the downstream activation of mitogen-activated protein kinase (MEK) and to the bypassing of the upstream BRAF inhibition [76]. Rac1 signaling has been identified as a major mediator of drug- and radio-resistance mechanisms [76]. NDV hyper-activation of Rac1 would overcome this resistance.(iii)The breaking of resistance to apoptosis. NDV was reported to have selectivity for apoptosis-resistant cells [77]. Infection could overcome the anti-apoptotic effect of the protein BcL-xL [77] and also that of the inhibitor of the apoptosis protein Livin [78].Another way to break resistance to apoptosis is to use a different way of programmed cell death. Ferroptosis is a new form of programmed cell death that is induced by NDV [56]. It involves the activation of p53 [79], downregulation of extracellular cystine uptake by xC^-^ antiporter [80], and iron-dependent accumulation of reactive oxygen species (ROS), leading to lethal levels of phospholipid peroxides. The recent study suggests that the p53-SLC7A11-PGX4 axis plays a central role in inducing ferroptosis, leading to a new form of NDV-induced cancer cell death [56].(iv)The breaking of resistance to hypoxia. Hypoxia inducible factor (HIF) is a transcription factor active in solid tumor microenvironments. It contributes to the tumor’s radio- and chemo-resistance. A velogenic NDV strain enhanced oncolytic activity via the MAPK pathway against a clear cell carcinoma line under hypoxic conditions [81].(v)The breaking of T cell tolerance towards TAA-expressing tumor cells. The NDV infection of human melanoma cells could break the tolerance of a melanoma-specific CD4+ T helper cell line [82].(vi)The breaking of resistance to TRAIL. Tumor necrosis factor (TNF)-related apoptosis-inducing ligand (TRAIL) resistant carcinoma lines were more susceptible to NDV-mediated oncolysis than TRAIL-sensitive cells. The IFN-stimulated gene (ISG)-12a enhanced the cells’ TRAIL sensitivity [83].(vii)The breaking of resistance to immune checkpoint blockade and to oncolysis. An intratumoral NDV application in B16 mouse melanoma could break systemic tumor resistance to immune checkpoint blockade immunotherapy [84]. A later study revealed that a similar effect was obtained with bladder cancer cells, even when employing oncolysis-resistant tumor cells [85].(viii)The breaking of resistance to anti-viral immunity. Pre-existing immunity to oncolytic NDV was reported to potentiate rather than to inhibit its immunotherapeutic efficacy [86]. This surprising result suggests that anti-viral immunity, considered as a major hurdle for effective therapeutic activity of OVs, is no hurdle for NDV.

Table 1 provides an overview of the milestone findings regarding the molecular determinants of the NDV-mediated anti-neoplastic effects.

**Summary**: Molecular mechanisms involved in NDV-mediated anti-neoplastic effects

-Cell infection by NDV can occur by different routes, e.g., membrane fusion, clathrin-mediated endocytosis, micropinocytosis, or RhoA-dependent endocytosis in lipid rafts (Rac1, TLR4).-Infected normal healthy cells prevent virus replication by establishing an interferon a/ß-induced anti-viral state.-NDV targets Rac1 upon endocytic entry of H-ras transformed cells. Rac1-Pak1 signaling, which is an important pathway for cancer cell migration, and tissue invasion is thus affected.-NDV exploits downregulated innate immunity pathways (PKR, RIG-I, IRFs) in cancer cells.-Tumor selectivity of viral replication involves: (i) the high cytoplasmic and cell surface expression of viral proteins, (ii) the induction of autophagy by viral proteins (NP or P), (iii) a switch to negative-strand antigenome synthesis, and (iv) a switch from the shut-off of cellular protein synthesis to the re-initiation of viral protein synthesis.-These steps enable virus replication, membrane budding with the help of the viral M protein, and virus release from tumor cells.-Tumor cell apoptosis is facilitated by eIF2a-CHOP-BcL-2/JNK and IRE1a-XBP1/JNK signaling and by microRNAs.-NDV spread in tumors is facilitated by secretory autophagy, syncytia, and NDV-related exosomes.-NDV can break cancer therapy resistances. The virus interferes with the Rac1 pathway, among others, which is of relevance to drug- and radio-resistance mechanisms.

## 4. NDV-Modified Cancer Vaccine for Cancer Immunotherapy

### 4.1. Successful Application of NDV for Antimetastatic Active-Specific Immunotherapy (ASI)

The prevention of metastatic spread was reported by postoperative active-specific immunotherapy (ASI) with NDV virally modified but not with unmodified irradiated autologous tumor cells in the ESb mouse tumor model [87]. This demonstrated a successful application of NDV for antimetastatic cancer immunotherapy. Associated studies in this model provided the earliest evidence for the generation of protective immune T-cell-mediated memory responses to cancer [88]. A recent review describes how cancer-reactive memory CD8+ T cell subsets orchestrate durable immunity to cancer [89].

These proof-of-principle ASI studies, performed from 1986 to 1990, could be reproduced in other animal models of metastasizing tumors. With regard to the mechanisms, it was found that the post-operative activation of tumor-specific cytotoxic T lymphocyte (CTL) precursors (CTLPs) from mice with metastases required stimulation with the specific tumor antigen plus additional signals. Such signals could be provided by NDV and resulted in the augmentation of CD4+ T helper (Th) and CD8+ CTLP T cell cooperation [90]. An essential NDV-mediated activation signal was provided by IFN-I [91].

In the period from 1990 to 2018, these studies were translated to human cells and a protocol was established for preparation of an NDV-modified autologous human tumor cell vaccine (ATV-NDV). This was then evaluated in a number of Phase I/II clinical post-operative ASI studies [92]. A prospective randomized–controled Phase II/III trial investigated the efficiency of ATV-NDV after liver resection for hepatic metastases of colon carcinoma patients as a tertiary prevention method. It revealed an improvement of long-term 10-year patient survival by as much as 30% [92]. The mechanisms of function, based on cancer-reactive memory T cells, just like in the pre-clinical studies, have been discussed [93].

### 4.2. Immunogenic Cancer Cell Death and Extrinsic Mechanisms of Oncolysis

The in situ activation of protective T cells can occur not only by post-operative immunization with a virus-modified cancer vaccine [87] but also by the direct inoculation of oncolytic NDV into primary tumors. Zamarin and colleagues demonstrated in 2014 that localized oncolytic virotherapy with NDV overcomes systemic tumor resistance to immune checkpoint blockade immunotherapy [84]. This was interpreted as being based on the induction of immunogenic cell death (ICD) and of systemic protective immunity leading to an abscopal effect at the site of a second untreated tumor.

Koks et al. [94] described in a murine orthotopic GBM model that NDV-mediated intratumoral virotherapy induces long-term survival and tumor-specific immune memory through the induction of ICD. The NDV infection of glioma cells induced the upregulation of DAMPs, such as calreticulin (Ecto-CRT), heat-shock proteins (HSP), high mobility group box 1 (HMGB1), and ATP [94]. The strong antigenicity of ICD-killed tumor cells was used for loading DCs and resulted in a vaccine with superior immunogenicity [95].

In another study, the evaluation of ultramicroscopic changes in GBM cells, upon infection by velogenic NDV, revealed a loss of tumor cells in all cell cycle phases (G1, S, and G2/M) accompanied with increases in the sub-G1 region (apoptotic bodies) [96].

Oncolysis by NDV is mediated by intrinsic pathways (see Section 3.7) and extrinsic (immune-system-mediated) pathways of cell death [97]. Extrinsic apoptosis can be induced by the HN protein. HN caused activation of spleen tyrosine kinase (Syk) and nuclear factor (NF) κB and upregulation of TRAIL in natural killer (NK) cells [98]. This led to the activation of caspase 8 and the cleavage of Bid into tBid, which transmits the extrinsic apoptotic signals to mitochondria [99]. NDV-activated macrophages produce nitric oxide (NO), which leads to mitochondrial lipid degradation, to the opening of mitochondrial permeability transition pores, and to the loss of mitochondrial membrane potential and the release of cytochrome c [99]. This then activates the intrinsic apoptosis pathway with the cleavage of caspase 9.

The early steps induced by the NDV infection of human tumor cells can be termed immunogenic apoptosis. In human MCF7 carcinoma cells, this involved the upregulation of the cell surface expression of viral HN and F proteins and also of HLA and cell adhesion molecules (ICAM-1, LFA-3) [38]. NDV also induces necroptosis in tumor cells. This is associated with the release of cytokines (IFN-I and tumor necrosis factor (TNF)-α) and chemokines (RANTES, IP-10) [38].

The pathogen-associated molecular patterns of NDV-infected tumor cells are (i) cytoplasmic ppp-RNA Leader [40] recognized by RIG-I, (ii) dsRNA recognized by PKR [32], and (iii) cell-surface-expressed HN protein recognized by NKp46 [100]. These PAMPs facilitate ICD.

### 4.3. Induction of Post-Oncolytic Immunity

The induction of post-oncolytic immunity by the intratumoral application of oncolytic NDV was analyzed in the above-mentioned orthotopic mouse glioma model in immunocompetent mice [94]. The therapeutic effects observed relied on the induction of ICD and on the induction of adaptive T-cell-mediated anti-tumor immunity. The NDV-treated tumors became infiltrated by T cells producing interferon gamma (IFN–γ). In immunodeficient T cell receptor V(D)J recombinase 2 knockout (RAG2 ^−/−^) mice or mice depleted of CD8+ T cells, no therapeutic effects were seen [94].

### 4.4. Inhibition of Cell Proliferation by IFN-I

Human type I interferons are a group of interferon proteins that regulate cell growth and help to regulate the activity of the immune system.

IFN-I and retinoic acid (RA) are known to inhibit the proliferation of many normal and transformed cells. Both have in vivo antitumor activity. Induction of the transcription factor (TF) IFN regulatory factor-1 (IRF-1) was reported to inhibit colorectal cancer proliferation and metastasis by suppressing the Ras-Rac1 pathway [101]. IRF-1 functions as a tumor suppressor. IRFs bind to the IFN promoters of specific target genes and induce the expression of IFN when tissues are infected.

### 4.5. NDV Induced Upregulation of MHC I

The NDV infection of human cancer cells leads to the upregulation of major histocompatibility complex (MHC) class I molecules (HLA) [38]. This can be explained by the cooperative interactions of two induced TFs, IRF-1, which binds to the interferon response sequence of the MHC I gene promoter, and NFκB, which binds to the respective enhancer region [102].

### 4.6. Viral Immune Escape Mechanisms

Viruses in their permissive hosts develop immune escape mechanisms that interfere with the IFN-I response and/or with proper MHC I-mediated antigen presentation. NDV in birds and Ebola virus in primates target and inhibit the IFN-I response [103]. Many human pathogenic viruses, such as herpesvirus, influenza virus, and HIV, target and downregulate MHC I-mediated antigen presentation to escape adaptive T-cell-mediated immune responses [104,105,106].

### 4.7. NDV-Induced Interferon Response: Inhibition of Virus Replication

IFN-I was discovered in 1957 by Isaacs and Lindenmann as a factor capable of interference with viruses [107]. Later, in 1974, Lindenmann suggested that viruses could be used as immunological adjuvants in cancer, based on his studies on viral oncolysis and post-oncolytic immunity [108]. In 1976, Gresser and colleagues demonstrated the importance of the early production of IFN-I in the response of the mouse to several viruses, including NDV [109].

Ohno and Taniguchi provided molecular evidence in 1983 for the presence of specific DNA sequences in the 5′-flanking region of the IFN-ß1 gene which were required for the NDV-mediated activation of the gene’s transcription [110].

IFN-ß proteins are produced in large quantities by fibroblasts while IFN-α proteins are produced in large quantities by plasmacytoid dendritic cells (pDCs). Chronic viral infection and cancer can suppress pDC-derived IFN-I [111].

### 4.8. NDV-Induced Interferon Response: Induction of an Anti-Viral State

In mammalian cells, NDV induces a strong type I interferon response [7,27,28,29]. This involves an early and a late phase and leads to the inhibition of virus replication.

In short, the early phase is initiated by cytoplasmic viral RNA-activating PKR and the RIG-I/mitochondrial antiviral signaling protein (MAVS) pathway. In the late phase, the released interferons α and ß initiate an amplification loop by activating IFNAR [29,30]. IFNAR stimulation rapidly triggers Janus kinase (JAK) and the signal transducer and activator of transcription (STAT) signaling cascades, which culminate via IRF-9/STAT1 TF complexes in the transcriptional regulation of hundreds of IFN-stimulated genes (ISGs). Transcriptional activation includes genes such as IFN-stimulated gene 15 (ISG15), CXCL10 chemokine IP10, IRF7, PKR, 2′-5′- oligoadenylate synthase (2′-5′-OAS), MxA, and others [27].

### 4.9. NDV Modified Dendritic Cell Vaccine IO-VAC^R^ and Individualized Multimodal Immunotherapy

In 2015, oncolytic NDV was successfully produced according to high-quality GMP criteria at IOZK. The institution also received a permit for its NDV-modified dendritic cell vaccine IO-VAC^R^ [112,113].

An approved individualized multimodal immunotherapy strategy (IMI) developed at IOZK was allowed to treat patients on a compassionate use basis [112,113]. A retrospective analysis of 70 GBM patients treated at IOZK revealed that a combination of first-line treatments with IMI resulted in a highly significant improvement in overall survival (OS) [114]. The combination of temozolomide with IMI also significantly improved the OS of GBM patients with very poor prognosis (IDH1 wild-type, MGMT promoter-unmethylated) [115].

## 5. Immune Cell Activation

When interferon was discovered, its function of virus interference was the first effect seen. Meanwhile, it is well established that type I IFNs exert direct effects not only on viruses but also on the cells of the immune system [116].

### 5.1. Activation of NK Cells, Monocytes, and Macrophages

The HN protein of NDV directly binds to the activating receptors NKp46 and NKp44 of murine NK cells. These two receptors facilitate signal transduction and NK cell activation via the immunoreceptor tyrosine-based activation motif (ITAM)-linked signaling chains CD3ζ (NKp46) and DAP12 (NKp44) [101]. The HN protein on infected cell membranes exists as a tetramer composed of a pair of dimers [117]. These multimer ligands lead to NK cell activation via cross-linking their activation receptors [118].

Activated NK cells can attack virus-infected cells and tumor cells and in theory also normal cells such as T cells. T cells lacking IFNAR are indeed highly susceptible to NK-cell-mediated killing. In the case of normal T cells, which do express IFNAR, however, IFN-I was reported to protect the cells against NK cell attack [119].

Human monocytes become activated upon contact with NDV. The tumoricidal activity of the activated monocytes is mediated through TRAIL [120]. NDV activates in murine macrophages NFκB and induces via inducible nitric oxide synthase (iNOS) [121] the production of NO [122] with its apoptotic effects via mitochondrial lipid degradation and cytochrome c release [99]. NDV-infected murine macrophages exert anti-tumor activity [122].

### 5.2. Dendritic Cell Activation

Of importance for the induction of adaptive T-cell-mediated immunity is the correct means of DC activation.

Human myeloid dendritic cells (mDCs) were infected by NDV to study a cellular response to virus infection that is not inhibited by any viral escape mechanism. A new approach of systems biology integrated genome-wide expression kinetics and time-dependent promoter analysis. Within 18 h, the cells established an anti-viral state. This was explained by a convergent regulatory network of transcription factors [TFs]. A network of 24 TFs was predicted to regulate 779 of the 1351 upregulated genes. The effect of NDV on mDCs was highly reproducible. The timing of this step-wise transcriptional signal propagation appeared as highly conserved [123].

The NDV infection of DCs activates NFκB, just like in NK cells and macrophages. This TF induces in DCs a module for the expression of pro-inflammatory cytokines together with a module for antigen presentation. These modules promote immunogenic DC1 differentiation with the potential to stimulate CD4+ T helper 1 responses and CD8+ effector T cells [124]. Cytokine production by DC1 is controlled by the upstream expression and action of a distinct transcriptional profile, including TFs, components of the antigen processing and cross-priming machinery and pattern recognition receptors [125]. Re-programmed in this way, immunogenic DC1 are the key drivers of adaptive immune responses directed against tumors with their TAs and against intracellular pathogens [125]. Immunogenic DC1 prime Th1 CD4+ T cells. Heightened expression of the TF IRF4 was reported to couple Th1 cell-fate determination with anabolic metabolism [126]. Other important factors for Th cell programming are the T cell receptor (TCR) and the CD28-mediated signal strength and activation of the Akt/mTOR pathway for induction of the TFs Tbet and Blimp-1 [127].

Antigen cross-presentation to CD8+ T cells is a specialized function conducted predominantly by DCs. Transporter associated with antigen processing (TAP) dysfunction in DCs enables noncanonical cross-presentation for T cell priming [128]. Oncolysate is internalized by DCs through either macropinocytosis or endocytosis. There is a spatial challenge of loading MHC I molecules from the ER with peptides from an extracellular source. It was recently reported that the receptor DNGR-1 expressed by DC1 binds dead-cell debris. Upon ligand engagement, the receptor DNGR-1 signals for phagosomal rupture to promote the cross-presentation of dead-cell-associated antigens [129]. TLR signals induce phagosomal MHC I delivery from the endosomal recycling compartment to allow cross-presentation [130]. DCs that had phagocytosed virus-infected but not uninfected apoptotic bodies activated cognate CD8+ T cells [125]. Such details are of relevance for understanding the functioning of the NDV oncolysate loaded DC vaccine IO-VAC^R^ from IOZK (Cologne, Germany) [113,114].

DCs pulsed with patient-derived NDV oncolysate were reported almost 20 years ago to potently stimulate autologous cancer-reactive memory T cells. Memory T cells from the bone marrow, activated in this way, secreted increased amounts of IFN-α and IL-15 [131] ligands which promote the survival, differentiation, and maintenance of CD8+ memory T cells [89].

DC maturation can be induced not only by native NDV but also by NDV-derived virus-like particles (VLPs). Such VLPs can induce DC maturation through the TLR4/NFκB pathway. This facilitates DC migration via the CCR7-CCL19/CCL21 chemokine axis [132]. The routes of internalization of the activated TLR4 include macropinocytosis, phagocytosis, and clathrin-mediated endocytosis, leading to TIR domain-containing adapter-inducing interferon ß (TRIF)-dependent signaling [133]. Lipid raft proteins such as CD14, CD44, HSP70, Lyn tyrosine kinase, and CD36 facilitate TLR4-mediated signaling [133].

### 5.3. Activation of T Cells

Naïve T cells are maintained in a quiescent state that promotes their survival and persistence [134]. Antigen recognition by naïve CD4+ and CD8+ T cells triggers mTOR activation which programs their differentiation into functionally distinct lineages [135]. Co-stimulatory and co-inhibitory receptors and ligands fine-tune various immune responses [136]. T cell signaling is also modulated by metabolic coordination [134] and by the actin cytoskeleton [137].

HN molecules at the surface of NDV-infected tumor cells introduce new cell adhesive strength for interaction with lymphocytes and for the costimulation of T cells [138]. HN tetramers with their multiple cell adhesion domains [9] and sialic acid binding sites [139] lead to the cross-linking of the corresponding cell surface receptors on T cells. This explains the observed production of IFN-I and TRAIL by HN but not F proteins on stimulator cells upon co-culture with human peripheral blood mononuclear cells [140].

Antigen peptide-presenting cells, stably transfected with HN c-DNA from NDV, caused a six-fold increase in peptide-specific CD8+ CTL responses, compared to untransfected antigen peptide presenting cells [141]. These findings from 1993 provided the first hints for the HN molecule’s T cell co-stimulatory activity.

The immunization of mice with NDV-infected tumor cells was demonstrated (i) to strongly increase tumor cell immunogenicity, (ii) to augment cooperative interactions between tumor-specific CD4+ Th and CD8+ CTLPs [90,91], and (iii) to increase the frequencies of CD4+ Th and CD8+ CTLPs via the induction of IFN-I [92].

Studies with human cells revealed that the NDV infection of tumor cells could break T-cell tolerance of a melanoma specific Th cell clone. This was due to the introduction of a CD80/CD86 independent co-stimulatory activity [82].

These examples demonstrate the T cell co-stimulatory activity of NDV for mouse CD8+T cells and for human CD4+ T cells. The increase in adhesive strength and costimulation could contribute to breaking T-cell tolerance to TAAs. T-cell tolerance towards TAAs and T-cell exhaustion are phenomena often observed in late-stage disease with a chronic excess of TAAs.

HN plasmid DNA was demonstrated to induce IFN-α and to be a powerful molecular adjuvant for anti-tumor vaccination [142]. Co-expression of a TAA and HN ensures the precise temporal and spatial co-delivery of an antigen and a co-stimulatory molecule, thereby potentiating anti-tumor immunity [143]. In general, immunotherapeutic gene delivery vectors should be applied at an optimal immunization site and should reach specific intracellular compartments, such as the cytosol or the nucleus [9]. An example is intranasal immunization with modified chitosan nanoparticles loaded with NDV DNA vaccine to enhance mucosal immune responses in chickens [144].

The above findings demonstrate that the HN molecules of NDV play an important role in the induction of an IFN-I response and that the IFN-I response in combination with an anti-TA response facilitates the induction and activation of a proper TAA-specific CD8+ CTL response, targeting the site of the tumor.

CTLs and NK cells eliminate infected cells or tumor cells by triggering apoptosis. The contents of lytic granules of killer cells, such as perforins [145] and granzymes [146], execute tumor-directed programmed cell death.

After the extensive proliferation and execution of effector programs within the first two weeks after antigen stimulation, a subset of these T cells becomes quiescent and develops into memory T cells [89]. The mechanisms of the differentiation of human memory T cells have been described following vaccination with live yellow fever virus (YFV) vaccine [147] and after vaccination with NDV-modified autologous tumor cell vaccine [94]. Long-lived specific CD8 T cells retained an epigenetic fingerprint of their effector history and remained capable of responding rapidly upon re-exposure to the YFV and also upon in vitro stimulation with a low dose of the homeostatic cytokine IL-15 [142]. Cancer-reactive memory T cells from breast cancer patients secreted increased amounts of IL-15 upon re-stimulation with NDV oncolysate-pulsed DCs in comparison to stimulation with DCs pulsed with oncolysate without NDV [131]. These studies suggest that NDV costimulation facilitates the secretion of homeostatic cytokines, which help maintain long-term T cell memory. In another study, tethered IL-15 augmented antitumor activity and promoted a stem-cell memory subset in tumor-specific T cells [148].

### 5.4. Oncolysis-Independent Immune Stimulatory Effects In Vivo

Virus potentiation of tumor vaccine T cell stimulatory capacity was reported to require cell surface binding but not infection [71]. This can explain why therapeutic effects in vivo can be obtained with NDV even against in vitro oncolysis-resistant tumor lines [85,149].

In a human melanoma nude mouse xenotransplant model, strong anti-tumor bystander effects were observed by a 20% co-admixture of melanoma cells pre-infected in vitro by lentogenic NDV (Ulster), suggesting oncolysis-independent innate immunity activation [150].

Oncolysis-independent effects are included in Table 1 with other anti-neoplastic effects. Table 2 provides an overview of the milestone findings with regard to NDV-mediated immune stimulatory effects.

**Summary:** Cellular and molecular mechanisms involved in NDV-mediated immune stimulatory activity

-Increase in adhesive interactions between infected tumor cells and immune cells (e.g., T cells and NK cells).-Activation of innate immunity cells: (i) signaling via HN-NKp44/46 in NK cells and (ii) NFkB mediated upregulation of TRAIL, secretion of TNFa and NO in monocytes and macrophages, and induction of a module for antigen presentation in DCs.-Reprogramming of DCs to DC1 by a choreographed cascade of transcription factors.-Oncolysate uptake by DCs and promotion of antigen cross-presentation.-Oncolysis-independent immune stimulation with pro-inflammatory and abscopal effects.-Augmentation of tumor cell immunogenicity.-Augmentation of CD8+ T cell costimulation.-Breakage of CD4+ T cell tolerance to tumor-associated antigens (TAAs).-Augmentation of cooperative interactions between CD4+ T helper and CD8+ cytolytic T cell precursors (CTLPs).-Increase in frequencies of CD4+ and CD8+ CTLPs via induction of interferon a/ß.-Induction of immunogenic cell death (ICD) with expression of PAMPs (HN, ppp-RNA Leader, and dsRNA) and release of DAMPs (ecto-CRT, HSP, HMGB1, and ATP).-Cognate interaction of DC1 TAA-presenting cells with CD4+ T cells leading to TA-specific activation and Th1 polarization.-Th1 CD4+ T cells interacting with CD8+ T cells helping their differentiation into CTLs and CD8+ memory T cells.-Patient-derived NDV-modified cancer vaccines (ATV-NDV and IO-VAC^R^) activating similar mechanisms in patients leading to long-term T cell mediated immune memory.

## 6. Schematic Diagram

Figure 1 shows a chart illustrating the mechanisms of the anti-tumor activity of oncolytic NDV. It represents a cycle of six steps in which the six viral proteins are involved. Mesogenic or velogenic NDV strains with their multicyclic replication capacity have the potential to drive several rounds of the cycle, thereby increasing the intensity and duration of the anti-cancer immune response.

## 7. Conclusions

As biological agents, OVs can replicate in tumor tissue and thereby augment their therapeutic effect. Tumor selectivity of this phenomenon ensures that healthy tissue is spared and that side effects are low.

This review reports milestones in the discovery of cellular and molecular mechanisms involved in tumor-selective virus replication and oncolysis and the resistance of normal healthy cells and of immune stimulatory effects.

It can be concluded that viral structural proteins from avian NDV successfully interact with proteins from non-permissive mammalian cells. This is not self-evident, considering the fact that the viral non-structural protein V antagonizes IFN α,ß signaling in cells from birds but not from humans. Avian NP interacts with mammalian eIF4E protein in viral mRNA translation and thus competes for binding with the corresponding cellular protein in the translation initiation complex. NP and P proteins induce autophagy in human cancer cells. M proteins interact during virus self-assembly with cellular MVB proteins. HN proteins interact with human NK-cell-activating receptors, co-stimulate T cells, and induce IFN-I responses. In addition, viral mRNAs interact with ribosomal proteins during protein translation, and ppp-RNA Leader interacts with RIG-I helicase. This latter interaction activates a cellular anti-viral defense signaling pathway. In contrast, NDV interacting with Rac1 GTPase in the lamellipodia of invasive cancer cells inactivates the cellular oncogenic signaling pathways [19].

Rac1 activates Pak1 signaling in non-infected tumor cells [20]. Rac1-Pak1 signaling affects tumor cell proliferation, migration, metastasis, angiogenesis, and drug- and radio-resistance [75,151]. The intrinsic GDP/GTP exchange activities of Rac1 are critical for mobilization of the actin filament system [21], leading to lamellipodia formation at the leading edge of migratory cells, including tissue invading cancer cells [151]. Oncolytic NDV is a biological agent targeting exactly this important oncogenic signaling pathway during endocytosis. It destroys tumor cells with invasive potential and alleviates therapy resistances.

Such resistance includes standard therapies depending on cell proliferation (chemo- and radiotherapy), as well as resistance to targeted therapies, apoptosis, TRAIL, and hypoxia. Of immunological relevance is that oncolytic NDV can break T-cell tolerance to TAs. This has to do with the HN-mediated improvement of adhesiveness between tumor cells and T cells and its co-stimulatory potential.

Other OVs also target oncogenic pathways. For example, activating mutations in the small GTPase Ras block PKR. This process can facilitate the selective replication of reovirus, HSV-1, adenovirus, and vaccinia virus [30].

NDV infection of cancer cells activates PI3K/Akt/mTOR, p38 MAPK/Mnk1, and CDK-Rb-E2F oncogenic pathways which help viral mRNA translation. Of importance for the understanding of tumor-selective virus replication and oncolysis are also the recent findings about the induction of ER stress and UPR signaling. NDV activates all three branches of UPR signaling: PERK-eIF2α, ATF6, and IRE1α. Signaling via these intrinsic pathways induces autophagy, apoptosis, and tumor-selective oncolysis. Distinct NDV proteins (NP, P, HN, and F) have been implicated in the tumor-selective induction of autophagy. Autophagy facilitates a switch from cellular to viral protein synthesis, antigenome self-assembly and viral replication in infected tumor cells. In addition to the intrinsic pathways of cell death, the induction of ICD causes activation of immune cells, which initiate the extrinsic mechanisms of cell death.

Structural studies of NDV viral proteins revealed various levels of assembly potential: NP monomers forming helical clam-shaped nucleocapsid structures and M proteins assembling in host cell plasma membranes into hollow helical oligomers; HN proteins on infected tumor cell membranes forming dimers and tetramers capable of interacting with NK cells and T cells, thereby cross-linking activating receptors; and molecular interactions between HN and F proteins, leading to fusion promotion activity of importance for virus cell entry, syncytium formation, and virus spread.

NDV can infect cells by membrane fusion, thus entering its genome directly into the cell’s cytoplasm where the viral RNA can activate PKR and RIG-I. In human non-transformed cells, such as lymphocytes, NDV induces in this way a strong IFN-I response, which prevents viral replication and leads to fast virus clearance. The strong IFN-I response protects infected cells in an autocrine way and uninfected cells in a paracrine way. While NDV and other virus-derived RNAs mediate crucial signals for innate immunity via the RIG-I-MAVS and IFNAR/JAK-STAT pathways, pathogen-derived DNA mediates respective signals via the cGAS-STING pathway [153].

In addition to its anti-neoplastic activities, NDV has immune stimulatory properties. The viral HN protein causes activation of NK cells by direct molecular interaction with the cytotoxicity-activating receptors NKp46 and NKp44. pHN plasmid DNA stimulates the production of IFN-I and can be used as a powerful vaccine adjuvant. Analysis of the induction requirements for T-cell-mediated protective anti-tumor immunity in an animal model revealed that type I IFNs play an important role in the augmentation of an anti-tumoral CD8+ CTL response.

Human dendritic cell activation by NDV can serve as a paradigm for the virus’s immune stimulatory properties. The cellular response is not inhibited by any viral escape mechanism. The establishment of an anti-viral state of DCs occurs within 18 h and involves in a choreographed way more than 24 TF proteins. One of these, NFκB, induces a module for antigen presentation and a module for the expression of pro-inflammatory cytokines for DC1 differentiation.

NDV oncolysate is taken up by DCs via macropinocytosis and via DNGR-1 receptor-mediated endocytosis. This leads to antigen processing and presentation via MHC II to CD4+ T cells and to antigen cross-presentation via MHC I to CD8+ T cells. Cooperative CD4+ and CD8+ T cell interactions, facilitated by NDV-induced IFN-I, then lead to the establishment of tumor-specific adaptive immune reactivity and systemic protective immunological memory.

Pre-clinical studies in animal models and progress in immunological basic research provide new insights into in vivo mechanisms [89,94]. For example, through the induction of ICD, intratumoral NDV virotherapy was demonstrated to induce long-term survival and tumor-specific immune memory [89] in an orthotopic mouse GBM model [95]. Postoperative active-specific immunotherapy studies in mice with an NDV-modified autologous tumor cell vaccine provided proof-of-principle for the induction of protective anti-tumor immunity and the prevention of metastatic spread [87,88].

Oncolytic NDV has been applied to cancer patients for more than 55 years and clinical effects have been described in several reviews [154,155,156,157]. One of the secrets of this success story is that this avian virus does not exert immune escape mechanisms in humans.

Evidence for the clinical utility and effectivity of NDV and NDV-derived cancer vaccines has been obtained in many experimental studies, including Phase I/II studies and one prospective randomized Phase II/III study. Further clinical trials are ongoing.

To conclude, OVs like NDV can target oncogenic pathways and exploit cancer immune evasion pathways and can induce local and systemic anti-tumor immunity. To perform such complex activities, oncolytic NDV makes use of all six of its structural proteins. Molecular analyses revealed that oncolytic NDV (i) can use multiple ways of cell entry, (ii) can interfere with multiple cancer-typical signaling pathways, (iii) can use multiple branches of UPR signaling, (iv) can kill cancer cells in multiple ways, (v) can use multiple ways of breaking therapy resistance, (vi) can activate immune cells in multiple ways, and (vii) can activate multiple interferon regulatory factors (IRF-1,-3,-4,-7,-9). Oncolytic NDV thus appears to have various options on how to infect and destroy human tumor cells. This explains the broad range of tumor cell types being susceptible to this OV: carcinoma, melanoma, GBM, lymphoma, etc.

With all the mentioned intrinsic activities, oncolytic NDV can be considered a potent platform for further modifications. One example of such a vaccine platform is a recently published NDV vector-based SARS-CoV-2 vaccine candidate. A study in 62 piglets revealed safety and immunogenicity upon intranasal application to induce mucosal immunity and upon intramuscular application. The vaccine can be produced at low cost in embryonated eggs in established facilities that are used to produce influenza virus vaccines [158].

Advances in the study of antitumor immunotherapy via NDV have recently been reviewed [31,157,159,160,161,162]. They manifest an increasing interest in the therapeutic potential of oncolytic NDV. The positive results obtained with NDV-modified cancer vaccine in colon cancer patients [93] and with NDV-modified dendritic cell vaccine in GBM patients [115] encourage future efforts in this direction.

Some of the oncolytic and immune stimulatory properties of NDV are shared by other OVs [1,30,159,161,162]. Examples of ICD-inducing OVs are measles virus, coxsackievirus B3, CD40-ligand-expressing adenovirus , parvovirus H-1, HSV, vaccinia virus, and reovirus [163]. With regard to oncolytic NDV there apparently exists pharmaceutical company interest [160].

## 8. Future Directions

So far, studies are lacking concerning the optimum delivery and dosage for maximal efficacy of oncolytic NDV. Further research is necessary to elucidate the mechanisms of NDV spread in tumors, to improve the efficacy of systemic tumor targeting, and to unravel the effects of NDV on cancer stem cells. Of particular interest is the combination of oncolytic NDV with immune checkpoint blockade therapy because synergistic effects are reported from pre-clinical studies.

OV therapy resistance is another problem of relevance. New biomarkers may help to elucidate molecular mechanisms of resistance and to predict OV therapy success [164]. Small molecule inhibitors (SMIs) could break OV therapy resistance in the case of vesicular stomatitis virus [164]. Vice versa, NDV could break resistance to SMIs [76].

Of great interest should be research to evaluate the potential of oncolytic NDV in targeting the many downstream pathways controlled by Rac1 and to compare this with SMI leads in cancer drug development [165]. The fact that NDV can break SMI resistance [76] and has low side effects should make this research effort worthwhile for commercial and patient benefit [166]. 

At times of personalized and individualized medicine the concept of randomized-controlled clinical trials (RCTs) as exerted by big pharma companies in the last decades is being questioned and should be further discussed [166]. A comparative analysis of the side effects of systemic therapies revealed that cancer vaccines and OVs exert profoundly lower side effects than agents such as cytostatic drugs and SMIs [167]. The aspect of side effects, such as major adverse events (WHO grades 3–4) should in future be considered more seriously in the evaluation of RCTs [167]. 

The therapeutic potential of NDV can be further augmented by the incorporation of therapeutic transgenes [160,168] and/or by the attachment of bispecific antibodies [169]. Engineering NDV with checkpoint inhibitors and immunocytokines could improve its effects upon intratumoral delivery [168]. Bispecific antibodies and trispecific immunocytokines could improve recruitment and activation of T cells and DCs to the site of cancer metastases [169].

Future improvements of clinical results in late-stage cancer patients can be expected by combining OV therapy with adoptive T cell therapy.

## 9. Article Highlights

Three main questions concerning oncolytic NDV and its mechanism of function were dealt with in this review. 1. How can the virus efficiently replicate in a tumor cell undergoing oncolytic cell death with a shut-off of cellular protein synthesis? 2. How can infected non-transformed human cells prevent virus replication? 3. How can different types of immune cells become activated by this avian paramyxovirus? According to latest and previous milestone findings the answers provided are: 1. Following NDV-induced PKR-eIF2α-mediated host protein shut-off in human cancer cells, the viral nucleoprotein NP binds to the cellular eIF4E protein within the mRNA translation initiation complex and directs translation exclusively towards viral proteins. In addition, the NP protein induces autophagy, protects the viral antigenome against degradation, and facilitates virus replication. 2. In non-transformed human cells, the signaling pathways induced by the interaction of viral pppRNA Leader with RIG-I and by dsRNA with PKR induce a strong IFN-I response, establish an antiviral state, and prevent viral antigenome nucleocapsid complex formation. 3. Innate immune cells become activated by NDV via NFκB and are re-programmed via distinct networks of transcription factors towards their specific effector functions. In the case of dendritic cells, this includes programs of antigen cross-presentation and of activation of pro-inflammatory cytokines. The viral HN protein activates NK cells via binding to NKp46 and activates IFN-I responses and co-stimulates T cells. Viral oncolysis causes release of TAs, PAMPs, DAMPs, and cytokines. This immune stimulatory environment leads to the generation of cancer-specific adaptive immune responses.

## Figures and Tables

**Figure 1 biomedicines-10-00562-f001:**
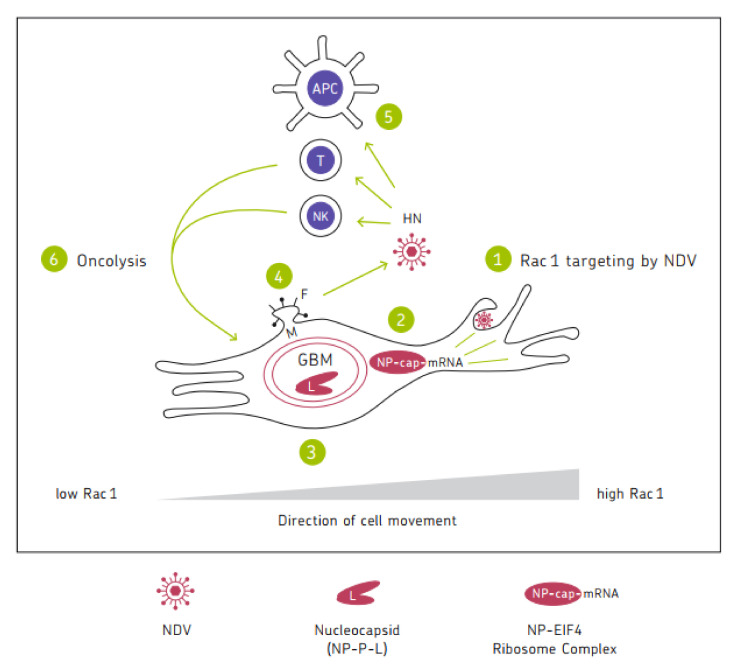
Schematic illustration of mechanisms of anti-tumor activity of NDV. This is exemplified with a migratory and invasive glioblastoma (GBM) cell. The direction of cell movement is accompanied by an increase in Rac1 expression from the trailing edge (**left**) to the leading edge at the lamellipodia (**right**) [151]. 1. Macropinocytosis/endocytosis of NDV-targeting Rac1, which plays a cardinal role in oncogenic alterations and in the development of drug resistance [152]. The direction of cell movement is accompanied by an increase in Rac1 expression from the trailing edge (**left**) to the leading edge at the lamellipodia (**right**) [151]. 2. Targeting the cap-dependent translational machinery; viral mRNA to protein translation in the cytosol and later in the double-membraned autophagosome (see GBM). This is achieved through the MNK1/2-eIF4E axis [35,36]. 3. Tumor-selective virus replication in autophagosomes. The clam shaped symbol stands for the helical nucleocapsid structure composed of L, P, and NP proteins [5,6]. 4. Virus progeny encapsulation, budding, and virus release mediated via M, HN, and F. 5. Antiviral response of healthy normal cells, including immune cells (APC, T, NK) initiated through recognition of HN by NKp46 and other PAMPs by TLRs and RLRs leading to IFN-I secretion and to DC1 and Th1 polarized adaptive immunity responses. 6. NDV-induced tumor cell death responses (oncolysis). These involve extrinsic (immune-mediated) and intrinsic cell death signaling pathways [98]. ICD-derived components feed into APCs which present TAAs to T cells. Several rounds of such cycles (1–6) drive oncolytic effects and lead to immunological memory and systemic antitumor immunity.

**Table 1 biomedicines-10-00562-t001:** NDV-mediated anti-neoplastic effects.

Feature	Mol Det	Year	Comment	Ref.
Tumor-selective Oncolysis	eIF2α-P eIF4E-NP	2006 2020	Shut-off of host cell protein Viral mRNA up	[32] [36]
	eIF2α-CHOP-BcL-2/JNK; NP and P IFNAR, EGFR, HN + F NLRP3 p53, xP^−^	2019 2016 2012, 2015 2004 2016 2021	UPR signaling; Autophagy; Viral prolif; Signaling down; Apoptosis; Pyroptosis; Ferroptosis	[45] [44] [29,30] [61] [41] [56]
Resistance of normal cells	PKR, RIG-I, IRF3,IFNß,IRF7 IFNAR TLR-3 TLR-4	2006 2009 2012 2019 2017	Kinetics and expression level high TRIF/IRF3 CD14/NFκB	[32] [28] [29] [16] [132]
	ppp-RNA Leader	2005 2017	Innate immunity activation; Stress granules	[39] [40]
Oncolysis-independent effects	HN; CAM; UV-NDV; costimulation	1997, 1998 2002 2010 1993	Augmented adhesiveness; T cell–tumor cell interaction	[72,138,140] [37] [142] [141]
	Augmented checkpoint inhibitory effects	2014 2018	Lysis-independent immune stimulation	[84] [85]
Syncytium formation	F and HN Exosomes	2004 2019	High cell surface expression; NP transfer	[61] [65]
	rVSV-NDV (F + HN)	2018	Platform for treatment of HCC	[63]
Breaking therapy resistances	costimulation Rac1; HIF; BcL-xL; Livin	2000 2010–2013 2018 2019	T cell tolerance; Radio- and chemotherapy; Apoptosis; Hypoxia	[82] [19,72,77,78] [77] [27]

eIF2α-P = phosphorylated eucaryotic translation initiation factor 2α; eIF4E-NP = viral nucleoprotein bound to this distinct protein of the eucaryotic translation initiation factor complex; CHOP = C/EBP-homologous protein; BCL2 = mitochondrial anti-apoptotic protein: JNK = c-Jun-N-terminal kinase; IFN = interferon; UPR = unfolded protein response; RIG-I = retinoic acid inducible gene I; IRF = interferon regulatory factor; ISG = interferon stimulated gene; CAM = Cell adhesion molecule, upregulated upon NDV infection of tumor cells; rVSV = recombinant Vesicular stomatitis virus vector; HCC = hepatocellular carcinoma; Rac1 = small Rho GTPase; HIF = hypoxia-inducible factor, an oxygen-sensitive transcription factor; BcL-xL = B-cell lymphoma-extra-large, an anti-apoptotic BcL-2 protein of the mitochondrial membrane; TRIF = TIR domain-containing adapter-inducing interferon ß, a necrosis pathway; TLR4 = lipopolysaccharide (LPS)-receptor-inducing pro-inflammatory signaling; NFkB = nuclear factor kappa B; Livin = a member of the human inhibitor of apoptosis proteins family; p53 = a tumor suppressor and significant player in normal and cancer immunity; xP^−^ = a membrane antiporter for uptake of extracellular cystin; GBM = Glioblastoma multiforme. Mol det = Molecular determinant.

**Table 2 biomedicines-10-00562-t002:** NDV mediated immune cell activation.

Feature	Mol Det	Year	Comment	Reference
NK cell activation	HN-NKp46	2009	NKpCD3-lacZ-inducible	[100]
	HN; TRAIL; Syk; NFκB	2017	IFN-γ independent	[103]
Macrophage activation	NFκB; NO	1996	In vitro	[122]
	IL1ß; NLRP3	2016	Human; Mouse	[41]
Monocyte activation	TRAIL; RIG-I; IRF7	2003	Viral replication not required	[120]
Dendritic cell activation	IFN-α; IL-15; IO-VAC^R^ DNGR-1 TLR	2002 2017 2020 2021 2014	Human MTC Reactivation Oncolysate cross-presentation	[131] [113] [125] [129] [130]
	24 TFs	2010	Uninhibited anti-viral response	[123]
	TLR4/NFkB CCR7	2017 2015	DC migration	[132] [133]
T cell activation	HN Perforins, Granzymes	1993 2020 2020	CD8+ CTL Pore formation Cytochrome c	[141] [145] [146]
	ATV-NDV	1989	CD4+ Th	[91]
	Costimulation	2000	Breaking T-cell tolerance	[82]
Breaking resistance to	TRAIL	2014	A role of ISG-12a	[83]
	Anti-viral immunity	2018	Anti-NDV immunity advantageous	[86]
	Immune checkpoint blockade	2014 2018	Abscopal effect on metastases	[84] [85]
ICD	TRAIL; ROS; IFN-I; TNF-α; RANTES; IP-10; Ecto-CRT; HSP; HMGB1; viral RNA, HN eIF2α-CHOP Cell-cell (NK-DC) interactions	2003 2002 2005 2017 2015 2019 2019	Immunogenic apoptosis and tumor cell necroptosis	[120] [37] [39] [40] [95] [45] [27]

Nkp46 = natural cytotoxicity receptor; NKpCD3-lacZ = a fusion protein composed of NKp46, the T cell receptor signaling chain CD3ζ and ß-galactosidase coded by the lacZ gene; Syk = spleen tyrosine kinase; NFκB = nuclear factor kappa B; NO = nitric oxide; NLRP3 = inflammasome, a multimeric cytosolic protein complex causing maturation of precursor forms of IL-1ß and IL-18 into active proinflammatory cytokines, thus mediating pyroptosis; TRAIL = tumor necrosis factor-related apoptosis-inducing ligand; ISG-12a = IFN-stimulated gene (ISG)-12a; RIG-I = retinoic acid inducible gene I; IRF = interferon regulatory factor; IL = interleukin; DC1 = a polarized dendritic cell; MTC = memory T cell; IO-VAC^R^ = NDV oncolysate pulsed dendritic cell vaccine; DNGR-1 = a receptor that binds dead cell debris and facilitates cross-presentation of corpse-associated antigens; TF = transcription factor; CCR7 = a chemokine receptor; VLP = virus-like particle; Th1 = CD4+ T helper 1 cell type; CTL = CD8+ cytotoxic T lymphocyte; ICD = immunogenic cell death; eIF2α-CHOP = see Table 1; TRAIL = tumor necrosis factor-related apoptosis-inducing ligand; ROS = reactive oxygen species; IFN-I = type I IFN; TNF-α = tumor necrosis factor α; RANTES = chemokine; IP-10 = chemokine; Ecto-CRT = plasma membrane expressed calreticulin; HSP = heat shock protein; HMGB1 = high mobility group box 1 protein; viral RNA = ppp-Leader and double-stranded (ds) RNA; HN = hemagglutinin-neuraminidase protein of NDV. Mol det = Molecular determinant.

## Data Availability

Not applicable.

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
