# Peer review of "Molecular Mechanisms of Anti-Neoplastic and Immune Stimulatory Properties of Oncolytic Newcastle Disease Virus"

_biomedicines, 2022, doi:10.3390/biomedicines10030562_

Round 1
Reviewer 1 Report
The review article entitled “Molecular mechanisms of anti-neoplastic and immune stimulatory properties of oncolytic Newcastle disease virus” submitted for publication in biomedicines is very comprehensive in scope. It is wonderful to have a review that covers how the virus infects target cells in birds and causes disease while infection of cells in humans, for example, does not cause illness. This is followed by a lengthy description of how the virus infects tumor cells and leaves them vulnerable to destruction by a healthy immune system. The article describes many studies performed with oncolytic NDV and the multitude of ways it leaves tumor cells vulnerable to anti-cancer treatments. Since the subject matter is very complicated it is good that the author does summaries of basic points throughout the article.
I have a few major suggestions and some minor corrections for the manuscript.
Major Points
- In your second paragraph (lines 33 – 39) you indicate that research with oncolytic NDV increased steadily from the 1950s, but near the end of the manuscript you indicate that Cassel et al. first reported the anti-neoplastic and immune-stimulatory properties of the virus in 1965. I think you should start the second paragraph with this information and then follow this up with a sentence beginning with “since this time”. The final sentence of the paragraph could indicate that you intend to describe and discuss the new findings in the present review.
- In section 3 for intrinsic and anti-neoplastic activities it would be helpful to provide a picture or drawing of some of the processes that you are describing in the sections. It is hard to follow due to the many abbreviations and factor names that are being mentioned. Therefore, a pictorial representation of what you are describing would be very helpful.
- In lines 371 and 372 the sentence beginning with Rac1 signaling is a little confusing. It might be clearer if you took out “targeted by NDV” and ended the sentence with “NDV hyper-activation of Rac1 would overcome this resistance too”.
- In lines 423 to 436 mention the years in which these studies were performed, since in line 437 you indicate that from 1990 to 2018 the results of the earlier studies were translated to human cells.
- I think that the review article should include this reference: Vijayakumar, G., McCroskery, S., and Palese, P. 2020. Engineering Newcastle disease virus as an oncolytic vector for intratumoral delivery of immune checkpoint inhibitors and immunocytokines. Journal of Virology 94(3): e01677-19.
Minor Points
- In line 30 place a comma after “existing virus”.
- In line 42 “bescribed” should be described.
- In line 93 delete spaces between L and [3,6]. Check the manuscript for other such instances.
- In line 129 change “human” to “humans”.
- In line 184 put a space between flow and cytometry.
- Define TA. I think this stands for tumor antigen, but I cannot find where this is stated in the review.
- In line 381 remove the word “and” then put a comma after ferroptosis.
- In line 470 change the word “loose” to “lose”.
- In line 522 add a reference for the information presented in the sentence.
- In line 540 change the word “allows” to “allowed” and correct the spelling for compassionate.
- The sentence in lines 556-557 is unclear. Try to restate it so the message is clearer.
- In line 565 change “right way of activation of DCs” to “correct means of DC activation”.
- In line 571 define TFs. I think this stands for transcription factors. It might have been defined earlier.
- In line 666 replace “maintenance of” with “maintain”.
- In line 671 replace “that” with “why”.
- In line 716 replace ‘outs” with “puts”.
- In line 724 change the sentence to “Rac1 activates Pak1 signaling in non-infected tumor cells”.
- In line 732 delete the words “that to”.
- In line 744 put “in infected tumor cells” at the end of the sentence.
Reviewer 2 Report
The review provides comprehensive information on modern data of molecular mechanisms of antineoplastic and immune stimulatory properties of the avian paramyxovirus Newcastle Disease Virus (NDV). Newcastle Disease Virus was one of the first oncolytic viruses to be discovered and has perhaps the longest history of medical use. Despite the fact that none of the NDV-based drugs has yet been approved for the clinical cancer treatment, this virus continues to be widely studied in the world. It is possible that the identification of detailed mechanisms of the antitumor activity of NDV will allow the development of an antitumor drug in the near future.
I have no major amendments for review, but there are some of editorial comments:
- The review is somewhat lacking in illustrative materials that would help to better understand the complex, multifaceted and very large amount of data on the molecular mechanisms of anti-neoplastic and immune stimulatory properties of Newcastle disease virus.
- It is necessary to unify the captions in the tables and it is desirable to remove the abbreviations like Mol det (Table 1,2), Viral prolif (Table 1).
Reviewer 3 Report
Dr. Schirrmacher has made extensive review on the topic. The molecular mechanisms of anti-tumoral and immunostimulatory properties by NDV has been discussed in depth. In most parts, it is well-organized.
One of the major limitations of this manuscript is that all discussions have been on NDV exclusively, even though a number of mechanisms and biology related to oncolytic properties for NDV are related or/and shared with other OVs. The inclusion of relevant information will enhance the quality of discussion, guide other investigators to pick right OV to apply to a particular type of cancer, and also potentially improve the citation of the article in the future.
For example, on page 6 (of 28), at the end of section 3.7., the author summarized the modes of cell death infected by NDV: “NDV induces multimodal cell death responses according to programmed cell death, including apoptosis [36], autophagic cell death [43], necroptosis [36], pyroptosis [54] and ferroptosis [55]”. To me, most of these types of cell death, belong to immunogenic cell death (ICD). As most other OVs can also elicit ICD, it is logic to make a statement on this fact, and probably cite one or two review article focusing on this discussion of ICD induced by various OVs. In fact, it has been the main topic in these reviews: (1). Oncolytic Immunotherapy: Dying the Right Way is a Key to Eliciting Potent Antitumor Immunity. Front Oncol. 2014;4:74. [PMID: 24782985]. (2). Oncolytic virotherapy and immunogenic cancer cell death: sharpening the sword for improved cancer treatment strategies. Mol Ther. 2014;22(2):251-256. [PMID: 24048442]
Minor issues:
- Page 1, lines 31-32. In fact, four, not three, oncolytic viruses have been approved. The fourth one is Delytact (teserpaturev), that was approved for the treatment of patients with malignant glioma in Japan in 2021.
- Page 3 of 28, lines 132. “NDV is likely the first OV with an identified oncogenic target molecule”. It is not. In fact, the human reovirus is an oncolytic virus that specifically targets cancer cells with an activated Ras pathway, as early as 1998. (Coffey MC, et al. Reovirus therapy of tumors with activated Ras pathway. 1998; 282:1332-4).
- The abbreviation “TAs” for tumor-associated antigens. I would suggest that “TAs” be changed to “TAAs” as the latter has been universally used by the peers in the field.
- Upon searching for ongoing clinical trials using NDV in cancer patients (clinicaltrials.gov), there is only one clinical trial ongoing (NCT04613492; phase 1). It seems that investigators are not interested in this particular OV as compared other OVs. What is your opinion on this?
- References:
(1). Ref #20: The year, volume, and article numbers are, 2014, 2014, 386470. (Missing the volume number that is 2014).
(2). Ref #65. The article number (missing) is, 527.
(3). Ref #71. This is author’s own article. The correct page numbers are, 1757-71; Not 1-15.
